# Asymptomatic malaria parasitaemia and seizure control in children with nodding syndrome; a cross-sectional study

Rodney Ogwang,[1,2] Ronald Anguzu,[1,2] Pamela Akun,[1,2] Albert Ningwa,[1,2] Edward Kayongo,[1] Kevin Marsh,[3] Charles R J C Newton,[4,5] Richard Idro[1,2,3]

[1]Makerere University College of Health Sciences, Kampala, Uganda
[2]Centre for Tropical Neuroscience, Kitgum site, Uganda
[3]Centre for Tropical Medicine and Global Health, University of Oxford, Oxford, UK
[4]Kenya Medical Research Institute-Wellcome Trust Collaborative Programme, Kilifi, Kenya
[5]Department of Psychiatry, University of Oxford, Oxford, UK

**Correspondence to**
Dr Richard Idro;
ridro1@gmail.com

## ABSTRACT

**Objective** *Plasmodium falciparum* is epileptogenic and in malaria endemic areas, is a leading cause of acute seizures. In these areas, asymptomatic infections are common but considered benign and so, are not treated. The effects of such infections on seizures in patients with epilepsy is unknown. This study examined the relationship between *P. falciparum* infection and seizure control in children with a unique epilepsy type, the nodding syndrome.

**Design** This cross-sectional study was nested in an ongoing trial 'Doxycycline for the treatment of nodding syndrome (NCT02850913)'. We hypothesised that, in patients with epilepsy, infection by *P. falciparum*, including asymptomatic infections, increases the risk of seizures and impairs seizure control.

**Setting and participants** Participants were Ugandan children with nodding syndrome, age ≥8 years, receiving sodium valproate. All had standardised testing including documentation of the number of seizures in the past month, a rapid malaria test and if positive, the peripheral blood parasite density.

**Outcomes** The primary outcome was the number of seizures in the past month (30 days).

**Results** A total of 164/240 (68%) had malaria. Asymptomatic infections (without fever) were seen in 160/240 (67%) and symptomatic infections in 4/240 (2.7%). In participants without malaria, the median (IQR) number of seizures in the past month was 2.0 (1.0–4.0) and it was 4.0 (2.0–7.5) in participants with malaria, p=0.017. The number of seizures in asymptomatic persons was 3.0 (IQR 2.0–7.3) and 6.0 (IQR 4.0–10.0) in symptomatic individuals, p=0.024. Additionally, in asymptomatic patients, a positive correlation was observed between the parasite density and number of seizures, r=0.33, p=0.002.

**Conclusion** In patients with nodding syndrome, both asymptomatic and symptomatic malaria are associated with an increased risk of seizures and poorer seizure control. Similar effects should be examined in other epilepsy disorders. Malaria prevention should be strengthened for these patients and chemotreatment and prevention studies considered to improve seizure control.

## Strengths and limitations of this study

► Nodding syndrome is a poorly understood epilepsy disorder not representative of other epilepsies. However, the disease offers the advantage of a uniform population of patients with epilepsy, receiving the same antiepileptic drug, and a similar level of care in Uganda.

► This was a cross-sectional study that cannot ascribe causality; prospective studies should be conducted to confirm the results.

► The study also relied on parental recall of the number of convulsive seizures in the past month and could have suffered from the shortfalls of recall bias. Again, a prospective determination of study outcomes will be more appropriate.

## INTRODUCTION

*Plasmodium falciparum* is still a major public health problem in tropical countries and especially, in sub-Saharan Africa. Over 200 million cases are reported annually with several thousand deaths majorly among children younger than 5 years and in pregnant women.[1] *P. falciparum* presents with a spectrum of manifestations from asymptomatic infections (malaria parasitaemia without fever), symptomatic but uncomplicated disease, to severe or complicated malaria.[2] Asymptomatic infections are common especially in highly endemic areas.[3–7] These symptomless malaria infections are generally considered benign and thought to be useful in maintaining immunity against severe disease.[8] Evidence is however emerging demonstrating that asymptomatic infections possibly have negative health effects (reviewed in Chen *et al* 2016)[9] including cognitive impairment,[10] anaemia,[11 12] co-infection with invasive bacterial disease[13] and increased maternal and neonatal mortality.[14]

About 50% of children with acute severe falciparum malaria present with neurological involvement.[15] *P. falciparum* is known to be epileptogenic, and is a leading cause of acute seizures in children living in malaria endemic areas.[16] However, the effects of asymptomatic infections on the incidence and control of

seizures in children with seizure disorders is unknown. This study examined the relationship between asymptomatic malarial infections and seizure control in patients with epilepsy using nodding syndrome as a model. The hypothesis is that: in patients with epilepsy, asymptomatic *P. falciparum* infections are associated with (1) Poorer seizure control. (2) Seizure control is worse in patients with higher parasitaemia.

Nodding syndrome is a poorly understood complex epilepsy disorder that affects children and adolescents in some regions of Africa.[17 18] Northern Uganda, South Sudan and southern Tanzania bear the greatest burden of this devastating disorder.[19 20] The aetiology is unknown but cross-reacting antibodies to *Onchocerca volvulus* have recently been proposed to underlie the pathogenesis.[18 21] Symptoms develop in previously normally developing children between the ages of 3 years and 18 years.[22 23] Patients present with a distinctive feature—clusters of head nodding—now defined as atonic seizures,[21] with a myoclonic element. The head nods present as repeated slow vertical head drops at a frequency of 5–20/min most often, on presentation of food or in cold weather.[24] Over time, the condition is complicated by multiple types of convulsive seizures (focal or multifocal, atypical absence, myoclonic jerks and generalised tonic-clonic seizures), behaviour difficulties and psychiatric disorders, cognitive decline, and in many severe cases, physical deformities and severe disability.[22] In Uganda, patients initiated on a specific symptomatic treatment intervention including the provision of sodium valproate as antiepileptic therapy obtained a 75% reduction in the burden of seizures.[25] The current study examined the relationship between asymptomatic *P. falciparum* infection and seizure control in nodding syndrome.

## METHODS
### Study design
This was a cross-sectional study of the relationship between asymptomatic *P. falciparum* malaria infections and seizure control in children with nodding syndrome. The study was nested within and all the participants are enrolled in an ongoing trial *'Doxycycline for the treatment of nodding syndrome (NCT02850913)'*. This trial is testing the hypothesis that nodding syndrome is an *O. volvulus* induced epileptic encephalopathy with antibodies to the parasite or its symbiotic bacteria, Wolbachia, cross-reacting with and damaging host neuron proteins.[18 26] Trial participants are randomised to either oral doxycycline 100 mg daily for 6 weeks or matching placebo.

### Setting
The trial is being conducted in the nodding syndrome affected districts of Kitgum, Pader and Lamwo in Northern Uganda. This region is inhabited by the Acholi, a Luo speaking community that is recovering from a decade-old civil war, with high levels of poverty and psychosocial problems. The districts are served by 17 nodding syndrome treatment centres where patients receive clinical care and treatment according to national guidelines.[27] The population prevalence of nodding syndrome in the affected age group in the region is 6.8 (95% CI 5.9 to 7.7) per 1000.[28] The region is also highly endemic to *P. falciparum* malaria and in 2015 and 2016 experienced a malaria epidemic.

### Study population
The study recruited all the 240 participants who had been enrolled in the Doxycycline for the Treatment of Nodding Syndrome Trial. The diagnosis of nodding syndrome was made according to the World Health Organisation (WHO) criteria. All were receiving sodium valproate (doses 12–35 mg/kg/day) as antiepileptic therapy plus nutritional, physical and psychological therapy.

### Procedures
#### Approvals
The Uganda National Council for Science and Technology and the National Drug Authority in Uganda provided regulatory approvals. Consent was obtained from each participant's carer and assent was obtained from the participants (except for cases with severe cognitive impairment).

### Screening, recruitment and clinical assessments
Most children with nodding syndrome in Uganda live within a few kilometres of the 17 nodding syndrome treatment centres in the country. Patient registers from the selected centres were accessed to identify potential participants. All patients with nodding syndrome in the specific locations were invited to the nearest follow-up centre or a central location in a village by the study field staff. Patients were then screened for eligibility and eligible participants consented. The inclusion criteria included participants with confirmed nodding syndrome as defined by WHO; age 8 years or older (to avoid doxycycline toxicity) and written consent by the parent or guardian. Women with a positive urinary Human chorionic gonadotrophin (HCG) (pregnancy) test, known hypersensitivity to tetracycline, reported inability to swallow capsules, enrolled into another trial and suspected high likelihood of non-compliance with the study drug and follow-up schedule were excluded.

As part of the requirements of the trial, all consenting participants and caregivers were invited to and hospitalised in the Kitgum General Hospital for about a week. During this period, they had detailed history, a full clinical and neurological assessment, an assessment of functioning using the Gross Motor Function Classification System and Modified Rankin Score, cognitive function on psychometric testing and the CogState (a computerised cognitive test), intellectual disability using the Child and Adolescent Intellectual Disability Screening Questionnaire, quality of life with the Quality of Life in Childhood Epilepsy Questionnaire and a diagnostic electroencephalogram test. The types of seizures were described and the burden reported as the number of seizures in the past month.

## Laboratory procedures

All participants had 10 ml of venous blood drawn for a complete blood count, liver and renal function, and study-specific tests. Malaria was tested using the *P. falciparum* malaria histidine rich protein-2 (HRP2) rapid diagnostic test (CareStart, 2016). Participants testing positive for malaria on the rapid diagnostic test (RDT) had thick Giemsa-stained blood smear slides prepared to determine the parasite density. Each slide was examined by two observers, and any differences were reconciled by a third observer. The number of asexual malaria parasites observed was reported per 200 white blood cells (WBCs) and the parasite density per microlitre of blood estimated assuming 8000 WBC/ul of blood. The laboratory technicians performing these tests were blind to the rest of the clinical information, including the burden of seizures in the past month, which was obtained earlier by the study clinical and nursing staff.

## Definitions and study outcomes

► Asymptomatic malaria was defined as *P. falciparum* parasitaemia with no history of fever in the past week and axillary temperature <37.5° C.

► Symptomatic malaria was defined as *P. falciparum* parasitaemia with either history of fever in the past week or axillary temperature ≥37.5°C.

► Seizure burden was defined as the number of seizures in the past 30 days as reported by the caretaker.

► Good seizure control was defined as No seizures in the past 30 days as reported by the caretaker.

► Poor seizure control was defined as one or more seizures in the past 30 days as reported by the caretaker.

In this report, only the relevant clinical and laboratory testing, obtained at the enrolment screening before the interventions, is reported.

## Patient and public involvement

The public was involved in developing the research questions of the overall study, that is, understanding the aetiology and treatment of nodding syndrome but not with the design of this substudy. The main trial is ongoing and in addition to personal contact with participants' families every 6 months, there are also community meetings every 3–6 months. The study results will be fed back to the community at these meetings and during the biannual local FM radio broadcasts that the study conducts.

## Data management and statistical analyses

Data were entered in a Microsoft Access Database using Epi info V.7.1.5.2. All patient data were then exported and analysed with STATA V.12.0 (StataCorp, Texas, USA) and GraphPad Prism V.6.01 (GraphPad Software, California, USA). Descriptive statistics where used to explore the data and this is reported as proportions, percentages, means (SD) and medians (IQR), as appropriate. Differences between the groups were tested by the $\chi^2$ test (proportions), the Student's t-test for normally distributed data, and the Mann-Whitney test was used for skewed data. Participants were categorised into those with good seizure control (no seizures in the past month) and those poor seizure control (one or more seizures in the past month). A logistic regression model was used to examine the relationship between seizure control and infection with Plasmodium. Variables with a p value of 0.3 or less at bivariate level were considered for multivariate

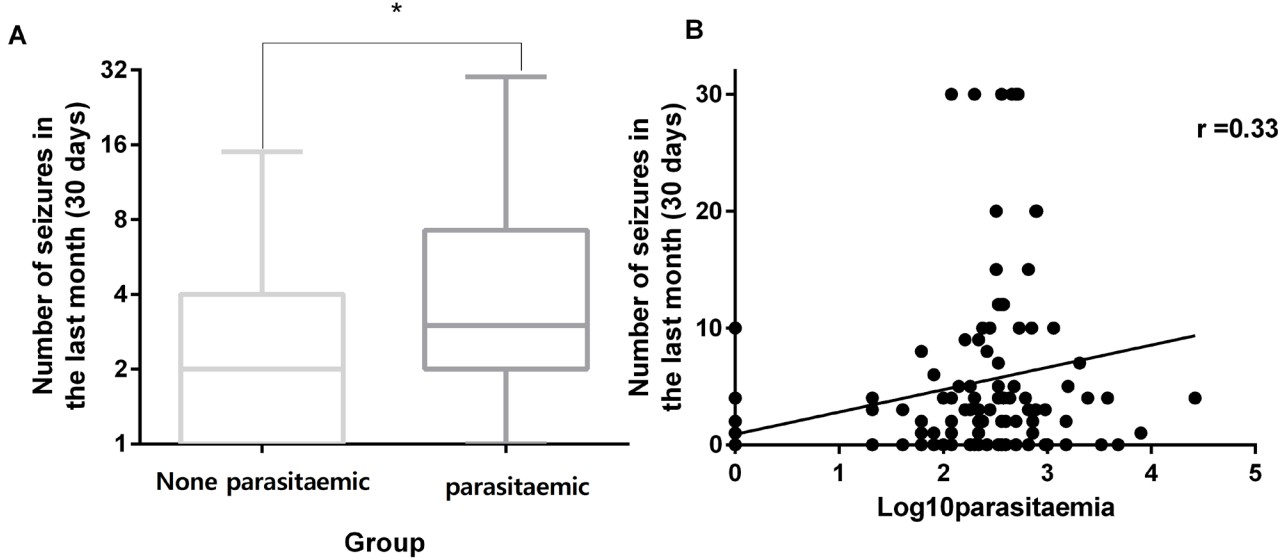

**Figure 1** (A) Compares the number (median) of convulsive seizures over the past month in patients with nodding syndrome with and without falciparum malatia parasitaemia. (B) The relationship between the parasite density and the number of seziures.

**Table 1** Summary of general characteristics of patients with nodding syndrome with and without malaria parastaemia

| | Patients without malaria parasitaemia, n=76 | Patients with malaria parasitaemia, n=160 | P values |
|---|---|---|---|
| Mean (SD) age, years | 15.8 (1.5) | 15.5 (2.1) | 0.22* |
| Sex, male (%) | 41 (54%) | 94 (58%) | 0.48† |
| Mean weight (SD), kg | 40.7 (9.8) | 41.9 (8.9) | 0.37* |
| Mean (SD) axillary temp, °C | 36.4 (0.42) | 36.5 (0.4) | 0.41* |
| Parasite density (IQR), /ul | 0 (0) | 200 (20–460) | <0.0001‡ |
| Mean (SD) dose of valproate acid, (mg/kg/day) | 24.1 (8.2) | 22.1 (6.3) | 0.06* |
| Median (IQR) number of clusters of head nodding episodes in the last month (IQR) | 3.0 (2.0–6.0) | 4.0 (2.0–10.0) | 0.38‡ |
| Type of seizures in the past month | | | |
| Absence, N (%) | 12 (15.7) | 20 (12.5) | 0.62† |
| Tonic, N (%) | 1 (1.3) | 0 (0.0) | 0.70† |
| Clonic, N (%) | 1 (1.3) | 1 (0.6) | 1.00† |
| Generalised tonic clonic, N (%) | 37 (50) | 73 (45.6) | 0.76† |
| Myoclonic, N (%) | 1 (1.3) | 1 (0.6) | 0.54† |
| Drop, N (%) | 2 (2.6) | 1 (0.6) | 0.24† |

*Unpaired students t test.
†$\chi^2$ test.
‡Mann-Whitney test.

analysis. Variables significantly associated with infection by *P. falciparum* were further assessed for interaction with Plasmodium infection and seizure control. All variables that were not in the model because of a non-significant p value were assessed for a confounding effect using a 10% difference between the adjusted and unadjusted odds to show confounding. To examine if there is a direct relationship between the malaria parasite density and seizure burden, Spearman's rank correlation testing was performed between the peripheral blood parasite density ($\log_{10}$ parasites/*ul*) and the number of seizures in the past month in patients with asymptomatic malaria.

## RESULTS
### General description
Between September 2016 and August 2017, a total of 240 patients with nodding syndrome was recruited into the trial. All were included in this study. Of these, 140/240 (58.3%) were male. The mean age was 15.6 (SD 2.0) years. The mean dose of sodium valproate was 22.6 (SD 7.2) mg/kg/day. A total of 164/240 (68.3%) participants tested positive for falciparum malaria, most of whom had asymptomatic infections (160/164 (97.6%)).

A total of 159/240 (66.3%) reported experiencing at least one convulsive seizure in the past year of whom 139 (57.9%) experienced at least one such seizure in the past 30 days. In patients without malaria, the median (IQR) number of seizures in the previous 30 days was 2.0 (1.0–4.0) and it was 4.0 (2.0–7.5) in patients with

malaria, p=0.017, Mann-Whitney test (figure 1). There were no significant differences in the doses (mg/kg/day) of sodium valproate across the groups. Among the four symptomatic malaria cases, the median (IQR) number of seizures in the past month was 6.0 (4.0–10.0). However, because these were very few, the group was excluded from all further analysis.

Among patients with asymptomatic malaria, the median (IQR) number of seizures experienced in the past month was 3.0 (2.0–7.3). Generalised tonic-clonic and absence seizures were the most common types of seizures described. Other seizure manifestations were infrequent. Also, there were no differences in the manifestations of seizures in the three groups (table 1).

### Relationship between other patient characteristics and seizure control in patients with nodding syndrome
Participants were categorised into those with good seizure control (no seizures in the past 30 days) and poor seizure control (one or more seizures in the past 30 days). In addition to presence of malaria parasitaemia, sex (AOR 1.96 (95% CI 1.11 to 3.46), p=0.03) and dose of antiepileptic drug (AOR 1.04 (95% CI 1.00 to 1.13), p=0.04) were significantly associated with seizure control (table 2). Furthermore, among children with asymptomatic malaria, a positive linear correlation was observed between the number of seizures in the past month and the peripheral blood parasite density r=0.33, (two-tailed p=0.002). A linear regression analysis gave the equation: Y=1.809X+2.549 (figure 1).

**Table 2** Factors associated with seizure control

| | Seizure control | | OR (CI) | P values | Adjusted odds | P values |
|---|---|---|---|---|---|---|
| | Good | Poor | | | | |
| Sex, n (%) | | | | | | |
| Male | 66 (48.15) | 71 (51) | | | | |
| Female | 34 (34.34) | 65 (65.6) | 1.77 (1.03 to 3.03) | 0.04 | 1.96 (1.11–3.46) | 0.02 |
| Age, mean (SD) | 15.69 (2.34) | 15.73 (1.66) | 1.01 (0.88 to 1.15) | 0.88 | | |
| Body temperature, mean (SD) | 36.46 (0.47) | 36.47 (0.45) | 0.98 (0.55 to 1.71 | 0.94 | | |
| Dose, mean (SD) | 23.42 (7.87) | 21.55 (5.88) | 1.04 (0.99 to 1.08) | 0.05 | 1.04 (1.00–1.13) | 0.04 |
| Weight, mean (SD) | 42.08 (9.35) | 41.5 (9.34) | 0.99 (0.966 to 1.02) | 0.6 | | |
| Duration with disease, mean (SD) | 8.44 (2.68) | 8.21 (3.07) | 1.02 (093 to 1.12) | 0.53 | | |
| $Log_{10}$parasitaemia, n (%) | | | | | | |
| 0 | 66 (53.3) | 57 (46.3) | | | | |
| 1 to 2.5 | 24 (38.71) | 39 (61.3) | 1.80 (0.93 to 3.36) | 0.06 | 1.81 (0.95–3.44) | 0.07 |
| >2.5 | 10 (20.0) | 40 (80.0) | 4.56 (2.09 to 9.9) | <0.01 | 5.11 (2.33–11.35) | <0.01 |

## DISCUSSION

This study investigated the relationship between asymptomatic malaria infection and seizure control in children with a complex epilepsy disorder, the nodding syndrome. The study found that patients with *P. falciparum* malaria infection experienced a significantly higher number of seizures in the previous month compared with patients without malaria and there was a direct correlation between the peripheral blood parasite load and the number of seizures. The study would suggest that in patients with seizure disorders on antiepileptic drug treatment, malaria parasitaemia, whether asymptomatic or symptomatic, may increase the risk of seizures and impair seizure control.

The association between acute infections and an increase in seizures in children with epilepsy is a well-recognised phenomenon. In the context of this study, *P. falciparum* is even thought to be epileptogenic.[29] In the malaria endemic regions of Africa, falciparum malaria is a leading cause of acute seizures and convulsive status epilepticus. This infection also contributes the largest fraction of seizure-related hospitalisations in children.[16 29 30] The seizures are not necessarily due to fever but are associated with increasing parasitaemia, possibly highlighting the pathological link between the presence of the parasites in the brain and the development of acute seizures.[29 31] Already, asymptomatic malaria infections have been shown to affect cognition in healthy children.[9] It was however not clear whether asymptomatic malaria parasitaemia would be associated with an increased frequency or severity of seizures. The finding of a dose-response effect of increasing seizures associated with higher parasite load and with recognised fever makes the association more compelling. To the best of our knowledge, this is the first study to demonstrate a relationship between asymptomatic malaria infections and a higher burden of seizures in children with epilepsy raising questions on

the meaning of 'asymptomatic malaria'. It may be that the symptoms of malaria do fall along a spectrum from mild to severe, with the fever reported by parents being an imperfect surrogate for pathophysiological disruption. Our study would therefore suggest that in children with epilepsy living in Africa, 'asymptomatic' malaria infections may not be truly benign but may be a risk factor impairing seizure control.

Apart from neurocysticercosis, the epileptogenic mechanisms of parasitic infections are not well understood.[32] *P. falciparum* infection is characterised by sequestration of the late stages of the intraerythrocytic cycle, particularly in the brain. In the case of severe malaria, acute seizures may potentially be induced through multiple pathways: (1) Indirectly through biochemical mechanisms associated with hypoglycaemia, hyponatraemia or acidosis.[29 31] (2) A direct effect of the parasites (or parasite toxin) sequestered in cerebral vessels.[29] (3) An immunological mechanism since high titres of voltage-gated cation channel antibodies have been observed in some children.[33] (4) Compromised perfusion of the brain due to cerebral microvascular parasite sequestration and raised intracranial pressure inhibiting substrate delivery.[29] (5) Downregulation of gamma-aminobutyric acid (GABA) receptors, thereby decreasing the inhibitory effects on seizures.[32] It is possible that, in patients with low seizure thresholds or a higher propensity to seizures such as children with epilepsy, even asymptomatic malaria infections may, by any of the above or other mechanisms, induce seizures and impair seizure control. Therefore, poor seizure control maybe an unrecognised consequence of asymptomatic malaria infections in children with epilepsy in malaria endemic regions.

Patients with malaria, on average, experienced twice the number of seizures experienced by those without malaria. The higher burden of seizures will impact on

the clinical and antiepileptic drug needs of the patients and on physician time, healthcare costs, patient's productivity, learning and achievements and on quality of life. There are an estimated 10 million people with epilepsy in Africa.[34] The prevalence of asymptomatic parasitaemia in school-age children in sub-Saharan Africa is 4%–64%.[35] Assuming an average 10% prevalence, there may be over one million people with epilepsy in Africa who are at risk of the potential adverse effects of 'asymptomatic' malaria-associated poorer seizure control. Should these findings be confirmed with other epilepsy disorders, it may be that patients with epilepsy in the malaria endemic areas of Africa should be considered a special or vulnerable group and considered for enhanced malaria prevention. Already, children with sickle cell anaemia living in similar settings are considered one such special group and in addition to the barrier methods of malaria prevention, such as bed nets, are offered enhanced malaria prevention through malaria chemoprophylaxis.[36] Thus, this study should be repeated in patients with other forms of epilepsy and if confirmed, trials should be conducted to evaluate if children with epilepsy may also benefit from malaria chemotreatment (of asymptomatic cases) and chemoprophylaxis against infection in addition to current barrier methods of malaria prevention.

The study had some limitations. First, nodding syndrome is a poorly understood complex epilepsy disorder that may not be representative of all other seizure disorders. However, the syndrome offers the advantage of a uniform population with all patients receiving the same antiepileptic drug and a similar level of care in Uganda.[27] Second, the study did not conduct more sensitive assays such as PCR to identify subpatent malaria infections. Third, there is the possibility of recall bias in the determination of the number of seizures in the past month. However, we limited this bias by focusing on convulsive seizures that are less likely forgotten, limited to the past month, collecting the information using standardised tools, and by clinicians trained and experienced in the care of patients with epilepsy. Lastly, this was a cross-sectional study with a limited sample size and so, a prospective study with a larger sample size should be conducted to confirm the results.

## CONCLUSION

In conclusion, in patients with nodding syndrome, both asymptomatic and symptomatic malaria infections are associated with an increased risk of seizures and poorer seizure control. Similar effects should be examined in patients with other epilepsy disorders. Malaria prevention should be strengthened for these patients and chemoprevention studies considered.

**Acknowledgements** RO thanks the IBRO: ARC 2017 Paper Writing Workshop, held in Entebbe, Uganda, for their mentorship. The authors thank the Doxycycline for the Treatment of Nodding Syndrome (DONS) trial implementation team; Nelson Odoch, Stephen Okiror, Innocent JJ Oryem and Deborah Akol.

**Contributors** RI and RO conceived and designed the study, carried out initial analyses and drafted the manuscript. RA, PA and AN designed the tools, collected data and critically reviewed the manuscript. EK performed the data analysis and critically reviewed the manuscript. KM and CRJCN designed the study and critically reviewed the manuscript.

**Funding** This study was jointly funded by the Medical Research Council (MRC) and the UK Department for International Development (DFID) under the MRC/DFID Concordat agreement through an African Research Leadership Award to RI and KM, grant number MR/M025489/1. The award is also part of the EDCTP2 programme supported by the European Union.

**Competing interests** None declared.

**Patient consent** Not required.

**Ethics approval** Makerere University School of Medicine Research and Ethics Committee and University of Oxford Tropical Research Ethics Committee .

**Provenance and peer review** Not commissioned; externally peer reviewed.

**Data sharing statement** A data sharing plan for the trial is being developed to have the overall study data available on public websites.

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
