## [Reviewer comments · BMJ Open]

ARTICLE DETAILS

TITLE (PROVISIONAL)	Asymptomatic malaria parasitaemia and seizure control in children with nodding syndrome; a cross-sectional study
AUTHORS	Ogwang, Rodney; Anguzu, Ronald; Akun, Pamela; Ningwa, Albert; Kayongo, Edward; Marsh, Kevin; Newton, Charles; Idro, Richard

VERSION 1 – REVIEW

REVIEWER	Janet Mifsud Department of Clinical Pharmacology and Therapeutics University of Malta Msida Malta
REVIEW RETURNED	10-May-2018

GENERAL COMMENTS	interesting paper, although some key details are missing. The data and results as presented are confusing and more data needs to be gender disaggregated as well as more information on effect of age. The cohort was one with a huge age variation from 5- 18 years and relevant information needs to be provided. otherwise good paper. The reviewer also provided a marked copy with additional comments. Please contact the publisher for full details.
---

REVIEWER	Scott Dowell BMGF, USA
REVIEW RETURNED	27-May-2018

GENERAL COMMENTS	This is a useful study documenting increased frequency of seizures amongst children with nodding syndrome who have malaria parasitemia compared with those who do not. The association between infection and increased seizures among children with epilepsy is well recognized, but it was not clear at the outset that so-called asymptomatic parasitemia would be associated with increased frequency or severity of seizures. The finding of a dose-response, with increasing seizures associated with higher parasite load and with recognized fever, makes the association more compelling. For the validity of this cross-sectional study it is important that the exposure variable (parasite load) and the outcome variable (frequency of seizures) were assessed by independent and appropriately blinded observers. The authors should clarify that those interviewing the parents about the frequency of the seizures were not aware of the parasite load, and that those quantifying the parasite load were not aware of the frequency of seizures. The novel contribution of the study rests on the fact that “asymptomatic infections” were associated with seizures, raising the question of the meaning of “asymptomatic”. The distinction between the 4 subjects with “symptomatic” infections (with reported or
---

	documented fever) and the 160 with “asymptomatic” infections (no reported fever in 7 days, and afebrile), may be somewhat arbitrary. Since the overwhelming majority of children in this population had parasitemia without reported fever, it seems more likely that the symptoms fall along a spectrum from mild to severe, with fever reported by parents an imperfect surrogate for pathophysiological disruption. The fact that seizures were increased even among those without apparent fever lends credence to this concept, but it would be useful for the authors to comment on the limitations of the “symptomatic/asymptomatic” distinction.
--	---

VERSION 1 – AUTHOR RESPONSE

Reviewer: 1

Interesting paper, although some key details are missing. The data and results as presented are confusing and more data needs to be gender disaggregated as well as more information on effect of age. The cohort was one with a huge age variation from 5- 18 years and relevant information needs to be provided. Otherwise good paper.

1. We have gone through the manuscript to make it clearer to the reader. Specifically, we have edited both the abstract and the methods sections to clearly show that this is a nested study within an ongoing study. The text in the methods section is to provide the context of the trial to the reader since the main study has not yet been concluded and to allow the reader to then focus on the specific focus of this study.

2. Regarding disaggregation by gender, we do not think this is necessary the burden and effects of malaria is similar in both gender i.e. malaria affects both gender with no preference.

3. As for age, as on page 7 under screening, recruitment and clinical assessments, we only recruited participants 8 – 18 years but not 5 – 18 years. In high malaria transmission areas like our study area, partial immunity to malaria is achieved by age 5 and so, the incidence of malaria beyond age 5 years is not expected to significantly vary by age group. Moreover, this study looked at the relationship between having malaria parasitaemia and the burden of seizures and even if this was to be influenced by age, such a relationship was examined and not seen. In table 2, there were no differences in the mean ages of children with ongoing seizures and those who had no seizures in the past month.

Reviewer 2

For the validity of this cross-sectional study, it is important that the exposure variable (parasite load) and the outcome variable (frequency of seizures) were assessed by independent and appropriately blinded observers. The authors should clarify that those interviewing the parents about the frequency of the seizures were not aware of the parasite load, and that those quantifying the parasite load were not aware of the frequency of seizures.

4. Thanks for this observation. We have now made this clarification on page 8. The screening and quantification of seizures was done before the lab tests and by two independent groups.

The novel contribution of the study rests on the fact that “asymptomatic infections” were associated with seizures, raising the question of the meaning of “asymptomatic”. The distinction between the 4 subjects with “symptomatic” infections (with reported or documented fever) and the 160 with “asymptomatic” infections (no reported fever in 7 days, and afebrile), may be somewhat arbitrary. Since the overwhelming majority of children in this population had parasitemia without reported fever, it seems more likely that the symptoms fall along a spectrum from mild to severe, with fever reported by parents an imperfect surrogate for pathophysiological disruption. The fact that seizures were increased even among those without apparent fever lends credence to this concept, but it would be

useful for the authors to comment on the limitations of the “symptomatic/asymptomatic” distinction. 5. Indeed this is the essence of the study. Is asymptomatic malaria really asymptomatic disease and with no risks? This discussion on pages 12 and 13 has now been revised extensively to include this observation.

VERSION 2 – REVIEW

REVIEWER	Janet Mifsud University of Malta
REVIEW RETURNED	06-Jul-2018
GENERAL COMMENTS	The authors have addressed well the concern and requests of the reviewers and the paper is now suitable for publication.